# Adaptive Feature Abstraction for Translating Video to Language

**Yunchen Pu** *
Department of Electrical and Computer Engineering
Duke University
yunchen.pu@duke.edu

**Martin Renqiang Min**
Machine Learning Group
NEC Laboratories America
renqiang@nec-labs.com

**Zhe Gan**
Department of Electrical and Computer Engineering
Duke University
zhe.gan@duke.edu

**Lawrence Carin**
Department of Electrical and Computer Engineering
Duke University
lcarin@duke.edu

## Abstract

A new model for video captioning is developed, using a deep three-dimensional Convolutional Neural Network (C3D) as an encoder for videos and a Recurrent Neural Network (RNN) as a decoder for captions. A novel attention mechanism with spatiotemporal alignment is employed to adaptively and sequentially focus on different layers of CNN features (levels of feature "abstraction"), as well as local spatiotemporal regions of the feature maps at each layer. The proposed approach is evaluated on the YouTube2Text benchmark. Experimental results demonstrate quantitatively the effectiveness of our proposed adaptive spatiotemporal feature abstraction for translating videos to sentences with rich semantic structures.

## 1 Introduction

Accurately understanding the fast-growing number of videos poses a significant challenge for computer vision and machine learning. An important component of video analyasis involves generating natural-language video descriptions, *i.e.*, video captioning. Inspired by the successful deployment of the encoder-decoder framework used in machine translation (Cho et al., 2014) and image caption generation (Vinyals et al., 2015; Pu et al., 2016; Gan et al., 2017), most recent work on video captioning (Venugopalan et al., 2015; Yu et al., 2016) employs a 2-dimentional (2D) or 3-dimentional (3D) Convolutional Neural Network (CNN) as an encoder, mapping an input video to a compact feature vector representation; a Recurrent Neural Network (RNN) is typically employed as a decoder, unrolling this feature vector to generate a sequence of words of arbitrary length.

Despite achieving encouraging successes in video captioning, previous models suffer from important limitations. First, the rich contents in an input video is often compressed to a single compact feature vector for caption generation; this approach is prone to miss detailed spatiotemporal information. Secondly, the video feature representations are typically extracted from the output of a CNN at a manually-selected fixed layer, which is incapable of modeling rich context-aware semantics that requires focusing on different abstraction levels of features. As investigated in Zeiler & Fergus (2014); Simonyan et al. (2014), the features from layers at or near the top of a CNN tends to focus on global semantic discriminative visual percepts, while low-layer feature provides more local, fine-grained information. It is desirable to select/weight features from different CNN layers

---

*Most of this work was done when the author was an intern at NEC Labs America.

adaptively when decoding a caption, selecting different levels of feature abstraction by sequentially emphasizing features from different CNN layers. In addition to focusing on features from different CNN layers, it is also desirable to emphasize local spatiotemporal regions in feature maps at particular layers.

To realize these desiderata, our proposed decoding process for generating a sequence of words dynamically emphasizes different levels (CNN layers) of 3D convolutional features, to model important coarse or fine-grained spatiotemporal structure. Additionally, the model employs different contexts and adaptively attends to different spatiotemporal locations of an input video. While some previous models use 2D CNN features to generate video representations, our model adopts the features from a pre-trained deep 3D convolutional neural network (C3D); such features have been shown to be natural and effective for video representations, action recognition and scene understanding (Tran et al., 2015) by learning the spatiotemporal features that can provide better appearance and motion information. In addition, the proposed model is inspired by the recent success of attention-based models that mimic human perception (Mnih et al., 2014; Xu et al., 2015).

The principal contributions of this paper are as follows: (*i*) A new video-caption-generation model is developed by dynamically modeling context-dependent feature abstractions; (*ii*) New attention mechanisms to adaptively and sequentially emphasize different levels of feature abstraction (CNN layers), while also imposing attention within local spatiotemporal regions of the feature maps at each layer are employed; (*iii*) 3D convolutional transformations are introduced to achieve spatiotemporal and semantic feature consistency across different layers; (*iv*) The proposed model achieves state-of-the-art performance on Youtube2Text benchmark. We call the proposed algorithm Adaptive SpatioTemporal representation with dynAmic abstRaction (ASTAR).

## 2 METHOD

Consider $N$ training videos, the $n$th of which is denoted $\mathbf{X}^{(n)}$, with associated caption $\mathbf{Y}^{(n)}$. The length-$T_n$ caption is represented $\mathbf{Y}^{(n)} = (\boldsymbol{y}_1^{(n)}, \ldots, \boldsymbol{y}_{T_n}^{(n)})$, with $\boldsymbol{y}_t^{(n)}$ a 1-of-$V$ ("one hot") encoding vector, with $V$ the size of the vocabulary.

For each video, the C3D feature extractor (Tran et al., 2015) produces a set of features $\mathbf{A}^{(n)} = \{\boldsymbol{a}_1^{(n)}, \ldots, \boldsymbol{a}_L^{(n)}, \boldsymbol{a}_{L+1}^{(n)}\}$, where $\{\boldsymbol{a}_1^{(n)}, \ldots, \boldsymbol{a}_L^{(n)}\}$ are feature maps extracted from $L$ convolutional layers, and the fully connected layer at the top, responsible for $\boldsymbol{a}_{L+1}^{(n)}$, assumes that the input video is of the same size for all videos. To account for variable-length videos, we employ mean pooling to the video clips, based on a window of length 16 (as in (Tran et al., 2015)) with an overlap of 8 frames.

### 2.1 CAPTION MODEL

For notational simplicity, henceforth we omit superscript $n$. The $t$-th word in a caption, $\boldsymbol{y}_t$, is mapped to an $M$-dimensional vector $\boldsymbol{w}_t = \mathbf{W}_e \boldsymbol{y}_t$, where $\mathbf{W}_e \in \mathbb{R}^{M \times V}$ is a learned word-embedding matrix, *i.e.*, $\boldsymbol{w}_t$ is a column of $\mathbf{W}_e$ chosen by the one-hot $\boldsymbol{y}_t$. The probability of caption $\mathbf{Y} = \{\boldsymbol{y}_t\}_{t=1,T}$ is defined as

$$p(\mathbf{Y}|\mathbf{A}) = p(\boldsymbol{y}_1|\mathbf{A})\prod_{t=2}^{T} p(\boldsymbol{y}_t|\boldsymbol{y}_{<t}, \mathbf{A}). \qquad (1)$$

Specifically, the first word $\boldsymbol{y}_1$ is drawn from $p(\boldsymbol{y}_1|\mathbf{A}) = \text{softmax}(\mathbf{V}\boldsymbol{h}_1)$, where $\boldsymbol{h}_1 = \tanh(\mathbf{C}\boldsymbol{a}_{L+1})$. Bias terms are omitted for simplicity throughout the paper. All the other words in the caption are then sequentially generated using an RNN, until the end-sentence symbol is generated. Conditional distribution $p(\boldsymbol{y}_t|\boldsymbol{y}_{<t}, \mathbf{A})$ is specified as softmax$(\mathbf{V}\boldsymbol{h}_t)$, where $\boldsymbol{h}_t$ is recursively updated as $\boldsymbol{h}_t = \mathcal{H}(\boldsymbol{w}_{t-1}, \boldsymbol{h}_{t-1}, \boldsymbol{z}_t)$. $\mathbf{V}$ is a matrix connecting the RNN hidden state to a softmax, for computing a distribution over words. $\boldsymbol{z}_t = \phi(\boldsymbol{h}_{t-1}, \boldsymbol{a}_1, \ldots, \boldsymbol{a}_L)$ is the context vector used in the attention mechanism, capturing the relevant visual features associated with the spatiotemporal attention (also weighting level of feature abstraction), as detailed in Sec. 2.2. The transition function $\mathcal{H}(\cdot)$ is implemented with Long Short-Term Memory (LSTM) (Hochreiter & Schmidhuber, 1997).

Given the video $\mathbf{X}$ (with features $\mathbf{A}$) and associated caption $\mathbf{Y}$, the objective function is the sum of the log-likelihood of the caption conditioned on the video representation:

$$\log p(\mathbf{Y}|\mathbf{A}) = \log p(\boldsymbol{y}_1|\mathbf{A}) + \sum_{t=2}^{T} \log p(\boldsymbol{y}_t|\boldsymbol{y}_{<t}, \mathbf{A}), \qquad (2)$$

Equation (2) is a function of all model parameters to be learned; they are not explicitly depicted in (2) for notational simplicity. Further, (2) corresponds to a single video-caption pair, and when training we sum over all such training pairs.

## 2.2 ATTENTION MECHANISM

We introduce two attention mechanisms when predicting word $y_t$: (i) spatiotemporal-localization attention, and (ii) abstraction-level attention; these, respectively, measure the relative importance of a particular spatiotemporal location and a particular CNN layer (feature abstraction) for producing $y_t$, based on the word-history information $y_{<t}$.

To achieve this, we seek to map $a_l \rightarrow \hat{a}_l$, where 4D tensors $\hat{a}_l$ all have the same dimensions, are embedded into same semantic spaces, and are aligned spatialtemporally. Specifically, $\hat{a}_l$, $l = 1, \ldots, L-1$ are aligned in the above ways with $a_L$. To achieve this, we filter each $a_l$, $l = 1, \ldots, L-1$, and then apply max-pooling; the filters seek semantic alignment of the features (including feature dimension), and the pooling is used to spatiotemporally align the features with $a_L$. Specifically, consider

$$\hat{a}_l = f(\textstyle\sum_{k=1}^{n_F^l} a_l(k) * \mathbf{U}_{k,l}), \tag{3}$$

for $l = 1, \ldots, L-1$, and with $\hat{a}_L = a_L$. $a_l(k)$ is the 3D feature map (tensor) for dictionary $k \in \{1, \ldots, n_F^l\}$ at layer $l$, and $\mathbf{U}_{k,l}$ is a 4D tensor. The convolution $*$ in (3) operates in the three shift dimensions, and $a_l(k) * \mathbf{U}_{k,l}$ manifests a 4D tensor. Function $f(\cdot)$ is an element-wise nonlinear activation function, followed by max pooling, with the pooling dimensions meant to realize final dimensions consistent with $a_L$. Consequently, $\hat{a}_{i,l} \in \mathbb{R}^{n_F^L}$ is a feature vector.

With $\{\hat{a}_l\}_{l=1,L}$ semantically and spatiotemporally aligned, we now seek to jointly quantify the value of a particular spatiotemporal region and a particular feature layer ("abstraction") for prediction of the next word. For each $\hat{a}_{i,l}$, the attention mechanism generates two positive weights, $\alpha_{ti}$ and $\beta_{tl}$, which measure the relative importance of location $i$ and layer $l$ for producing $y_t$ based $y_{<t}$. Attention weights $\alpha_{ti}$ and $\beta_{tl}$ and context vector $z_t$ are computed as

$$e_{ti} = \boldsymbol{w}_\alpha^T \tanh(\mathbf{W}_{a\alpha}\hat{a}_i + \mathbf{W}_{h\alpha}\boldsymbol{h}_{t-1}), \quad \alpha_{ti} = \mathrm{softmax}(\{e_{ti}\}), \quad \boldsymbol{s}_t = \textstyle\sum_{l=1}^L \alpha_{ti}\hat{a}_i, \tag{4}$$

$$b_{tl} = \boldsymbol{w}_\beta^T \tanh(\mathbf{W}_{s\beta}\boldsymbol{s}_{tl} + \mathbf{W}_{h\beta}\boldsymbol{h}_{t-1}), \quad \beta_{tl} = \mathrm{softmax}(\{b_{tl}\}), \quad \boldsymbol{z}_t = \textstyle\sum_{l=1}^L \beta_{tl}\boldsymbol{s}_{tl}, \tag{5}$$

where $\hat{a}_i$ is a vector composed by stacking $\{\hat{a}_{i,l}\}_{l=1,L}$ (all features at position $i$). $e_{ti}$ and $b_{tl}$ are scalars reflecting the importance of spatiotemporal region $i$ and layer $t$ to predicting $y_t$, while $\alpha_{ti}$ and $\beta_{tl}$ are *relative* weights of this importance, reflected by the softmax output. In (4) we provide attention in the spatiotemporal dimensions, with that spatiotemporal attention shared across all $L$ (now aligned) CNN layers. In (5) the attention is further refined, focusing attention in the layer dimension.

## 3 EXPERIMENTS

We present results on Microsoft Research Video Description Corpus (YouTube2Text) (Chen & Dolan, 2011). The Youtube2Text contains 1970 Youtube clips, and each video is annotated with around 40 sentences. For fair comparison, we used the same splits as provided in Yu et al. (2016), with 1200 videos for training, 100 videos for validation, and 670 videos for testing. We convert all captions to lower case and remove the punctuation, yielding vocabulary sizes $V = 12594$.

We consider the RGB frames of videos as input, and all videos are resized to $112 \times 112$ spatially, with 2 frames per second. The C3D (Tran et al., 2015) is pretrained on Sports-1M dataset Karpathy et al. (2014), consisting of 1.1 million sports videos belonging to 487 categories. We extract the features from four convolutional layers and one fully connected layer, named as *pool2, pool3, pool4, pool5* and *fc-7* in the C3D (Tran et al., 2015), respectively.

Table 1: Results on BLEU-4, METEOR and CIDEr metrics compared to state-of-the-art results (Yu et al., 2016) on Youtube2Text. respectively.

| Methods | BLEU-4 | METEOR | CIDEr |
|---------|--------|--------|-------|
| h-RNN [4] | 49.9 | 32.6 | 65.8 |
| ASTAR | **51.74** | **36.39** | **72.18** |

The widely used BLEU (Papineni et al., 2002), METEOR (Banerjee & Lavie, 2005) and CIDEr (Vedantam et al., 2015) metrics are employed to quantitatively evaluate the performance of our video caption generation model, and other models in the literature.

Results are summarized in Tables 1, and we outperform the previous state-of-the-art result on Youtube2Text. This demonstrates the importance of leveraging intermediate convolutional layer features. In addition, we achieve these results using a single model, without averaging over an ensemble of such models.

## 4 CONCLUSION AND FUTURE WORK

We have proposed a novel video captioning model, that adaptively selects/weights the feature abstraction (CNN layer), as well as the location within a layer-dependent feature map. Our model achieves state-of-the-art video caption generation performance on Youtube2Text benchmark.

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
