# Peer review of "Adaptive Feature Abstraction for Translating Video to Language"

_ICLR 2017 — rejected_

[Public Comment · (anonymous) · 22 Nov 2016 (modified: 05 Dec 2016)]
**missing citations in related work**

You should also probably cite

[Official Review · AnonReviewer2 · rating 4 · confidence 4 · 16 Dec 2016]
**solid work but lack of novelty**

The authors apply the image captioning architecture of Xu et al. 2015 to video captioning. The model is extended to have attention over multiple layers of the ConvNet instead of just a single layer. Experiments on YouTube2Text, M-VAD and MSR-VTT show that this works better than only using one of the layers at a time.

I think this is solid work on the level of a well-executed course project or a workshop paper. The model makes sense, it is adequately described, and the experiments show that attending over multiple layers works better than attending over any one layer in isolation. Unfortunately, I don't think there is enough to get excited about here from a technical perspective and it's not clear what value the paper brings to the community. Other aspects of the paper, such as including the hard attention component, don't seem to add to the paper but take up space. 

If the authors want to contribute a detailed, focused exploration of multi-level features this could become a more valuable paper, but in that case I would expect a much more thorough exploration of the choices and tradeoffs of different schemes without too many spurious aspects such as video features, hard attention, etc.

[Official Review · AnonReviewer1 · rating 7 · confidence 4 · 16 Dec 2016]
**Clear and efficient space+time+feature attention mechanisms for video captioning**

1) Summary

This paper proposes a video captioning model based on a 3D (space+time) convnet (C3D) encoder and a LSTM decoder. The authors investigate the benefits of using attention mechanisms operating both at the spatio-temporal and layer (feature abstraction) levels.

2) Contributions

+ Well motivated and implemented attention mechanism to handle the different shapes of C3D feature maps (along space, time, and feature dimensions).
+ Convincing quantitative and qualitative experiments on three challenging datasets (Youtube2Text, M-VAD, MSR-VTT) showing clearly the benefit of the proposed attention mechanisms.
+ Interesting comparison of soft vs hard attention showing a slight performance advantage for the (simpler) soft attention mechanism in this case.

3) Suggestions for improvement

Hypercolumns comparison:
As mentioned during pre-review questions, it would be interesting to compare to the hypercolumns of

[Official Review · AnonReviewer3 · rating 4 · confidence 5 · 17 Dec 2016]
**state-of-the-art results but too incremental**

This paper presents a model for video captioning with both soft and hard attention, using a C3D network for the encoder and a RNN for the decoder. Experiments are presented on YouTube2Text, M-VAD, and MSR-VTT. While the ideas of image captioning with soft and hard attention, and video captioning with soft attention, have already been demonstrated in previous work, the main contribution here is the specific architecture and attention over different layers of the CNN.

The work is well presented and the experiments clearly show the benefit of attention over multiple layers. However, in light of previous work in captioning, the contribution and resulting insights is too incremental for a conference paper at ICLR. Further experiments and analysis of the main contribution would strengthen the paper, but I would recommend resubmission to a more suitable venue.

[Author Response · Martin Renqiang Min · 20 Jan 2017]
**Authors' final rebuttal**

We thank all the reviewers for the critical comments.

All the reviewers agree that our experimental results convincingly support our hypothesis. The disagreement is that whether our paper is suitable for ICLR and whether our contribution is important to know by the representation learning community.

We made the following changes in the pdf file revision addressing the reviewers’ concerns:

1. We updated the abstract, emphasizing that the goal of our paper is not only about presenting another architecture for video captioning, but also about new video representations that are suitable for the video captioning task. Attention mechanisms are just technical means to achieve this goal.

2. We updated the introduction, adding three technical challenges to overcome to use adaptive spatiotemporal feature representations with dynamic feature abstraction for the captioning task. We also explained why naïve approaches such as MLP/max(average)-pooling did not meet our requirements.

3. We updated the related work including the hypercolumn representation.

4. We put additional experimental results comparing our approach ASTAR to hypercolumns, MLP, and max/average-pooling in Table 1 on Page 8. Our results clearly demonstrate the significance of dynamically selecting a specific level. The hypercolumn representation without level selection has much worse performance than our method.

5. We updated Figure 1 and moved Figure 2 below it to emphasize our contributions following the review comments.

6. We added the reference of Yao et al., ICCV 2015.

In summary, we believe that our contributions are clear and important to know by the representation learning community. This line of thinking might influence other researchers to perform additional research on classification and other tasks. It also inspires us to design new deep architectures to efficiently learn and effectively utilize different levels of feature representations in a dynamic fashion.

[Final Decision · Program Chairs · 06 Feb 2017]
**ICLR committee final decision**

Reviewers feel the work is well executed and that the model makes sense, but two of the reviewers were not convinced that the proposed method contains enough novelty in light of prior work. The comparison of the soft vs hard attention model variations is perhaps one of the more novel aspects of the work; however, the degree of novelty within these formulations and the insights obtained from their comparison were not perceived as being enough to warrant higher ratings. We would like to invite the authors to submit this paper to the workshop track.